# Pilot study of large-scale production of mutant pigs by ENU mutagenesis

Tang Hai[1,2†], Chunwei Cao[1,2†], Haitao Shang[2,3†], Weiwei Guo[4†], Yanshuang Mu[2,5†], Shulin Yang[2,6], Ying Zhang[1,2], Qiantao Zheng[1,2], Tao Zhang[1,2], Xianlong Wang[1,2], Yu Liu[2,3], Qingran Kong[2,5], Kui Li[2,6], Dayu Wang[1,2], Meng Qi[1,2], Qianlong Hong[1,2], Rui Zhang[1,2], Xiupeng Wang[1,2], Qitao Jia[1,2], Xiao Wang[1,2], Guosong Qin[1,2], Yongshun Li[1,2], Ailing Luo[1,2], Weiwu Jin[1,2], Jing Yao[1,2], Jiaojiao Huang[1,2], Hongyong Zhang[1,2], Menghua Li[1,2], Xiangmo Xie[1,2], Xuejuan Zheng[1,2], Kenan Guo[2,3], Qinghua Wang[2,3], Shibin Zhang[2,3], Liang Li[2,3], Fei Xie[2,3], Yu Zhang[2,5], Xiaogang Weng[2,5], Zhi Yin[2,5], Kui Hu[2,5], Yimei Cong[2,5], Peng Zheng[2,5], Hailong Zou[2,5], Leilei Xin[2,6], Jihan Xia[2,6], Jinxue Ruan[2,6], Hegang Li[2,6], Weiming Zhao[2,6], Jing Yuan[2,6], Zizhan Liu[2,6], Weiwang Gu[2,7], Ming Li[2,7], Yong Wang[2,3], Hongmei Wang[1,8*], Shiming Yang[4,8*], Zhonghua Liu[5,8*], Hong Wei[3,8*], Jianguo Zhao[1,8*], Qi Zhou[1,8*], Anming Meng[8,9*]

[1]State Key Laboratory of Stem Cell and Reproductive Biology, Institute of Zoology, Chinese Academy of Sciences, Beijing, China; [2]Chinese Swine Mutagenesis Consortium Working Group, Chinese Swine Mutagenesis Consortium, Beijing, China; [3]Department of Laboratory Animal Science, College of Basic Medicine, Third Military Medical University, Chongqing, China; [4]Department of Otolaryngology-Head and Neck Surgery, Institute of Otolaryngology, Chinese PLA General Hospital, Beijing, China; [5]College of Life Science, Northeast Agricultural University of China, Harbin, China; [6]Institute of Animal Sciences, Chinese Academy of Agricultural Sciences, Beijing, China; [7]Pearl Laboratory Animal Sci. & Tech. Co. Ltd, Guangzhou, China; [8]Chinese Swine Mutagenesis Consortium Guide Group, Chinese Swine Mutagenesis Consortium, Beijing, China; [9]School of Life Sciences, Tsinghua University, Beijing, China

*For correspondence: wanghm@ioz.ac.cn (HWa); yangsm301@263.net (SYan); liu086@126.com (ZLi); weihong63528@163.com (HWe); zhaojg@ioz.ac.cn (JZ); qzhou@ioz.ac.cn (QZho); mengam@mail.tsinghua.edu.cn (AM)

[†]These authors contributed equally to this work

Competing interests: The authors declare that no competing interests exist.

**Abstract** N-ethyl-N-nitrosourea (ENU) mutagenesis is a powerful tool to generate mutants on a large scale efficiently, and to discover genes with novel functions at the whole-genome level in *Caenorhabditis elegans,* flies, zebrafish and mice, but it has never been tried in large model animals. We describe a successful systematic three-generation ENU mutagenesis screening in pigs with the establishment of the Chinese Swine Mutagenesis Consortium. A total of 6,770 G1 and 6,800 G3 pigs were screened, 36 dominant and 91 recessive novel pig families with various phenotypes were established. The causative mutations in 10 mutant families were further mapped. As examples, the mutation of *SOX10* (R109W) in pig causes inner ear malfunctions and mimics human Mondini dysplasia, and upregulated expression of *FBXO32* is associated with congenital splay legs. This study demonstrates the feasibility of artificial random mutagenesis in pigs and opens an avenue for generating a reservoir of mutants for agricultural production and biomedical research.

## Introduction

Pigs are an important source of meat worldwide and are widely used in biomedical research (*Groenen et al., 2012*; *Prather, 2013*; *Wernersson et al., 2005*). The similarities of organ size, anatomic and physiologic characteristics and genome sequence between pigs and humans stimulated the use of these animals as preferred models of various human diseases and as sources of allogenetic organs for xenotransplantation. Genetic modifications in pigs are crucial for the generation of tailored disease models, and the knowledge-based development of sustainable pig production relies on the annotation of porcine genomes and the precise and efficient generation of tools for genetic engineering. Although a draft of pig genome was reported in 2012 and much progress has been made in pig gene identification, mapping and functional analyses, the current knowledge of the pig functional genome remains limited (*Brown and Moore, 2012*). Understanding the pig genome and identifying the causative genes for certain physiologies are imperative and will not only benefit disease modeling and xenotransplantation but also be invaluable for solving a range of problems associated with pork production (*McGonigle and Ruggeri, 2014*).

ENU-induced mutagenesis is an effective forward genetic approach for identifying functional genes and generating animal models. ENU is a potent mutagen that primarily induces point mutations and chromosome rearrangements in the genome in a random manner (*Russell et al., 1979*). The advantages of this approach are that no assumptions are needed regarding the underlying genetic causes of physiologic or biologic processes, and it allows the precise selection of phenotypes of interest (*Acevedo-Arozena et al., 2008*). ENU mutagenesis has advantages over knockout strategies in that it generates a series of point mutations that frequently mimic the subtlety and heterogeneity of human genetic lesions (*Oliver and Davies, 2012*). In the past, ENU mutagenesis had been used to generate thousands of mutants in *Caenorhabditis elegans* (*De Stasio and Dorman, 2001*), flies (*Choi et al., 2009*; *Yu et al., 1987*), zebrafish (*Driever et al., 1996*; *Geisler et al., 2007*; *Haffter et al., 1996*; *Wienholds et al., 2003*; *Xiao et al., 2005*) and mice (*Hrabé de Angelis et al., 2000*; *Nolan et al., 2000*; *Vitaterna et al., 1994*), leading to the discovery of many important genes and genetic pathways and contributing much to the current understanding of embryonic development, organogenesis and the etiology of various diseases. However, the effectiveness of ENU mutagenesis in large mammalian species remains unknown. The need to annotate functional genomics and to generate ideal large animal models prompted us to conduct a forward genetic screen using ENU in pigs.

Here, we report the first evidence of the feasibility of the ENU mutagenesis screen in large animals to generate inheritable mutants primarily focused on dysmorphology, growth rate, body weight and blood biochemical parameters. A three-generation breeding scheme was employed for mutation screening, and 36 dominant and 91 recessive novel pig lines were identified. We show that the ENU-induced mutations in various lines can be mapped. These findings revealed that ENU mutagenesis in pigs is an efficient strategy to identify genes with novel functions and to generate mutants for agricultural production and biomedical research.

## Results

### ENU mutagenesis in Bama miniature pigs

The Bama miniature pig, a southern Chinese native breed with a body weight of 20–30 kg at 6 months of age, was used for mutagenesis in the present study. To determine an appropriate mutagenic but non-toxic dose of ENU (*Justice et al., 2000*), an ENU concentration of 65 or 85 mg/kg bodyweight was intravenously injected three times, with a week between the injections, into 7- to 8-month-old boars (G0) (*Figure 1A*) (*Hrabé de Angelis et al., 2000*). The treated boars showed poor-quality sperm at 2–3 weeks after the last injection and underwent a period of azoospermia for 6–8 weeks. G0 boars recovered fertility at 12–14 weeks post-treatment, and there were no obvious differences in sperm quality between the two treatment groups (*Figure 1—figure supplement 1*).

To evaluate the frequency and mutational pattern of ENU-induced nucleotide changes in pigs, we implemented a trio-based method to assess the ENU-induced mutation frequencies on a genome-wide scale. Two G0 boars, one injected with 65 mg/kg ENU and the other with 85 mg/kg ENU, were selected, and each of these G0 boars produced five G1 progenies, each of which derived from a specific ENU mutagenized sperm of G0. Using trio-based 2b-RAD sequencing, the G1-boar-specific

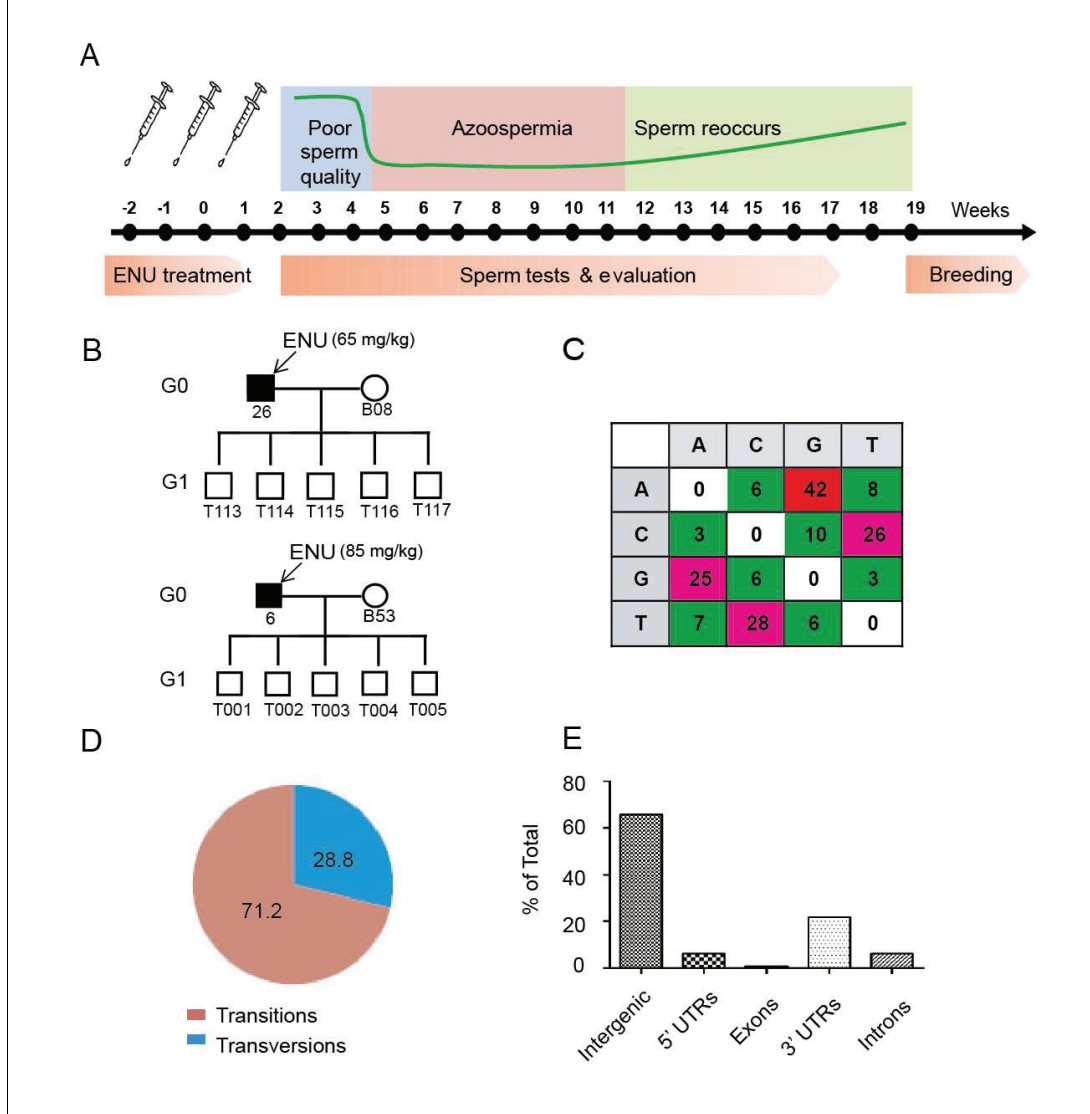

**Figure 1.** Protocol used for ENU treatment and estimation of mutation spectrum by 2b-RAD sequencing. (**A**) A scheme for ENU treatment. (**B**) Pedigrees of two families treated with different doses of ENU (65 and 85 mg/kg) were selected for analysis. Each family consisted of an ENU-treated G0 boar, an untreated sow, and five G1 boars. (**C**) Spectrum of ENU-induced mutations revealed by RAD sequencing. ENU treatment predominantly introduced G>A transitions, which are marked in red. (**D**) Transversion mutations occurred at a frequency of 28.8%, whereas transition mutations occurred at a frequency of 71.2%. (**E**) Most induced mutations were located in intergenic and intron regions, whereas only a small percentage of mutations resided in the genomic regions containing exons, 5' UTRs and 3' UTRs.

The following source data and figure supplement are available for figure 1:

**Source data 1.** ENU-induced mutations revealed by RAD sequencing.
**Figure supplement 1.** Sperm quality assessment in pigs pre- and after ENU treatment.

mutations, which were absent from the G0 boars and G0 sows, were identified as the ENU-induced mutations. The results indicated that ENU treatment produced a significant number of G to A transitions, and these transitions occurred far more frequently than did transversions (71.2% for transitions versus 28.8% for transversions) (*Figure 1B,C and D*). An increase of mutation frequency in the 85 mg/kg treatment group was observed compared with the 65 mg/kg treatment group ($5.86 \times 10^{-6}$ vs. $1.63 \times 10^{-6}$) (*Supplementary file 1* and *Supplementary file 2*). The mutation frequency

indicated that each G1 pig carried 4,500–16,458 heterozygous mutations throughout the genome (pig genome size $2.81 \times 10^9$ bp), most of which were presumably located in the intergenic area (*Figure 1E*). An archive of DNA, tissues, cells and sperm from the 5,000 G1 offspring was established to harbor the mutations. We concluded that a dose of $3 \times 85$ mg/kg ENU for three treatments is suitable for effective mutagenesis in Bama miniature boars.

## Systematic phenotypic screening for dominant and recessive mutants

A three-generation breeding scheme was designed to screen dominant and recessive mutations (*Figure 2A*). G0 boars were mated with wild-type sows to produce G1 founders, which were observed for novel phenotypes caused by dominant mutations. At the same time, G1 individuals without detectable mutant phenotypes were randomly selected for further breeding to detect recessive mutant phenotypes in G3. In detail, one G1 boar was mated with 3–4 sows to produce G2 offspring, and at least four G2 sows were backcrossed with the G1 father to generate G3 offspring. Considering the dual purpose of swine in agriculture and biomedical research, a specific phenotyping pipeline was established to screen G3 offspring for recessive mutants. The phenotyping pipeline was primarily focused on the following postnatal abnormalities: coat color changes; congenital malformations; abnormal blood biochemical parameters; behavior abnormalities; and growth traits, including growth rate, body weight and body size (*Figure 2B*). All pigs underwent the listed tests in an order designed to minimize carry-over influences. To identify heritable traits, the phenotypes that were reproducible in G1 and G3 in independent pedigrees were identified and further bred for inheritance analysis. In this pilot study, 36 dominant mutant lines were recovered from the screening of 6,770 G1 offspring, indicating a mutation frequency of ~0.27% (*Table 1*). Among these offspring, seven mutant pigs were confirmed to have traits that had dominant inheritance (19%, 7/36) (*Table 1*, *Figure 3A–G*). Four hundred G1 boars that showed no detectable abnormal phenotype were gradually introduced into the mating pedigrees for recessive mutation screening (*Figure 2A*), and 6–10 litters with not less than 30 G3 progenies for each G1 pedigree were generated for each G1 founder. A total of 169 pedigrees produced more than 30 G3 offspring, and 91 mutant lines were identified, giving a rate of approximately 0.5 mutants (91/169) per pedigree, which is similar to that observed in mice (*Hrabé de Angelis et al., 2000*). Among these mutants, 22 of the 91 mutant lines were confirmed as inheritable. Inheritance testing and screening for additional recessive mutations are still ongoing.

The identified mutants exhibited a broad range of phenotypes, including hearing loss, skeletal defects, coat color changes, hind-limb paralysis, chapped skin, increased growth rate and body size change. These mutants could thus potentially be utilized to develop models of human diseases and to decipher the underlying mechanism of phenotypes controled by pig genetics (*Table 1*).

## Description of several mutant lines and mapping of mutations

To detect the ENU-induced mutations in mutant lines, we developed a family-based genome-wide linkage study (GWLS) in combination with whole-genome sequencing (WGS). To date, we have successfully mapped causative regions or mutations in 10 different mutant lines (*Table 2*). The line Z0009 showed ectropion, flattening of the ears and large, thick, plate-like scales over the entire body (*Figure 3J*), similar to the characteristics of congenital harlequin ichthyosis (OMIM, 242500), and linkage analysis identified its mutant locus at chr15: 120–134 Mb interval. Lines Z0078 and Z0079 (*Figure 3M*), which had a significant linkage signal at chr13: 210–216 Mb, showed abnormal facial and limb development similar to characteristics observed in popliteal pterygium syndrome (OMIM, 263650). These mutation lines provided potential large animal models to aid in determining the pathomechanisms of rare human diseases.

When compared with wild-type pigs, homozygous mutant pigs of line Z0006 showed a significantly higher body weight at 6 months of age (33.95 ± 7.39 kg vs. 20.04 ± 5.68 kg in wild type pigs (WT), p=3.37E-06) and an increase in daily body weight gain (0.23 ± 0.03 kg/d vs. 0.18 ± 0.04 kg/d in WT), phenotypes that were inherited in a recessive manner (*Figure 3N*). The genome-wide association study revealed a strong association between this phenotype and the chr6: 122–123 Mb region (p=4.00E-06). Three other strains (Z0139, Z0154 and Z0156) showed decreased body size and weight at 6 months of age (10.60 ± 2.19, 10.02 ± 3.82, 11.94 ± 1.56 vs. 20.99 ± 1.32 in WT all p<0.05) and

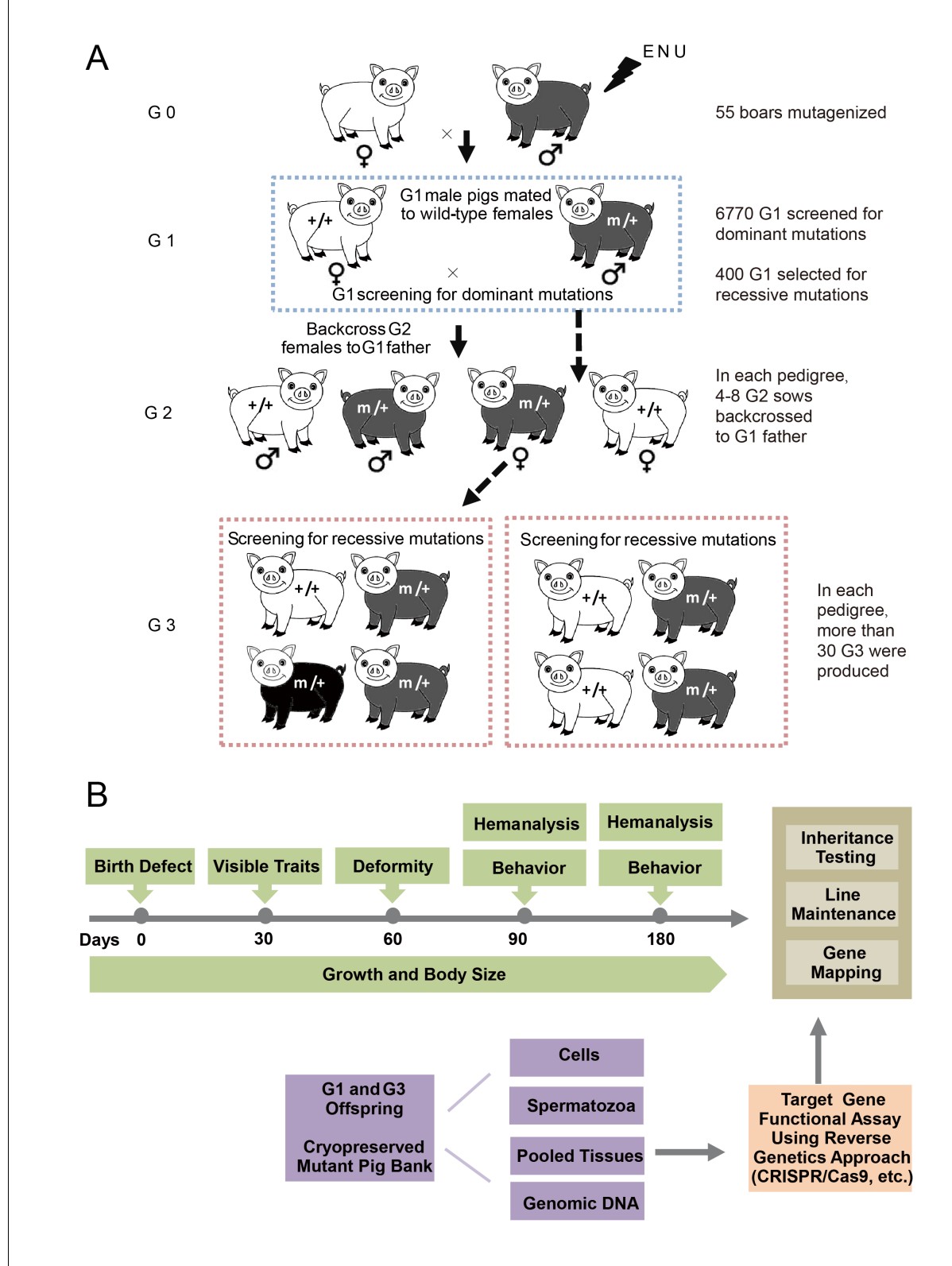

**Figure 2.** The breeding scheme used to screen dominant and recessive mutations and a phenotype screen pipeline, and the establishment of the mutant database in pig ENU mutagenesis. (**A**) A three-generation breeding scheme for mutation screening. ENU treatment-induced mutations were transmitted to the G1 generation. The G1 offspring that were heterozygous for ENU-induced mutations were screened for aberrant phenotypes. The G1 boars were mated to wild-type sows to produce G2 offspring. The G2 daughters were mated back to their father, and recessive mutations were

*Figure 2 continued on next page*

*Figure 2 continued*

then detected in the G3 progenies. (**B**) To minimize the number of pigs used in the mutant screen, offspring bred from the ENU-treated individuals were screened by different methods. Sperm, DNA, tissues and cells were archived from the G1 offspring.

could therefore potentially be bred as miniaturized experimental pigs with an adult body weight of less than 10 kg at adulthood.

Coat color phenotypes have been extensively selected through pig domestication and affect consumer selection in some countries (*Hirooka et al., 2002*). The 'Liang-tou-wu' coat color pattern (black coat on the head and bottom and white coat on the body) is a unique pattern in certain Chinese native breeds, and its genetic mechanism remains unclear. The current study has identified several coat color mutants, including white coat color (Z1202101, *Figure 3A*; TBB007T095, *Figure 3G*), single-end coat color (Z0017, Z0022 and Z0040; *Figure 3H*), diluted coat color (Z0037), diluted brown coat color (Z0015, *Figure 3I*), and black coat color (A0049, *Figure 3E*). Interestingly, three mutant pedigrees (Z0017, Z0022 and Z0040) that displayed the 'single-end black' coat color phenotype were derived from different G0 boars (*Table 2*). These mutants could be used to decipher the molecular mechanisms that underlie coat color formation and to breed experimental miniature pigs with white coats. Furthermore, coat-color-related mutants may be models of neurodegenerative disorders and cancers because of their physiologic characteristics. *Xu et al. (2010)* reported that a mutation in *Archain 1* resulted in a diluted coat color and Purkinje cell degeneration. The occurrence of cutaneous melanoma, a malignant cancer, is strongly associated with mutations in pigmentation-related genes; thus, coat color mutant pigs could be effective models for studying melanomas (*Eggermont et al., 2014*).

To date, by combining family-based GWLS and WGS, we have successfully mapped causative regions or mutations in 10 different mutant lines (*Table 2*, *Figure 3G–O*, Figure 5A). This work supports the validity of the number of mutations in the pig genome induced by an appropriate ENU treatment protocol in the present study.

## A mutation in *SOX10* causes inner ear malfunctions and mimics human Mondini dysplasia

The mutant line (TBB007T095) was characterized first by white coat color and diluted iris pigmentation, which showed a dominant inheritance pattern (47%, 63/134) (*Figure 3G*, *Figure 4A*). All individuals of the mutant family were then assessed for auditory function based on an evoked auditory brainstem response (ABR) as hypopigmentation is often associated with hearing loss (*Chen et al., 2016*). Our results showed that all mutants with a white coat color showed profound hearing loss, and no peak could be provoked in response to stimulus intensities until 120 dB SPL, whereas wild-type individuals had hearing thresholds of 30–40 dB SPL (*Figure 4A*). GWLS under a dominant model revealed only one genome-wide significant linkage signal (LOD >3) in chromosome 5 (7–71 Mb) (*Figure 4B*, *Figure 4—figure supplement 1*). To isolate the causative gene, the hypopigmentation-related gene *SOX10* within the linkage region was first selected as a candidate gene, and a missense mutation (c. 325 A>T) that resulted in a highly conserved amino acid substitution (R109W) was identified by sequencing the *SOX10* coding sequence (*Figure 4B* and *Figure 4C*). Furthermore, whole-genome sequencing of two independent mutants and one wild-type pig with average read-depth coverages of 11.51×, 10.64×, and 10.63×, respectively, was performed to investigate the mutations in the linkage region. The results indicated that only the *SOX10* R109W mutation ($SOX10^{+/R109W}$), but no other variants, completely co-segregated with the hearing-loss phenotype (*Figure 4—figure supplement 2*). Furthermore, this mutation was not observed in the dbSNP database (https://www.ncbi.nlm.nih.gov/projects/SNP/) or in other pig breeds (*Supplementary file 3*), indicating that this mutation was generated by ENU mutagenesis. The genotyping results of the *SOX10* c. 325 A>T mutation in members of whole mutant families further supported the evidence showing that *SOX10* c. 325 A>T is responsible for the mutant phenotype (*Figure 4D* and *Supplementary file 3*).

The inner ear of the wild-type miniature pig presents several key features that are similar to those of the human ear (*Guo et al., 2015*). Human hearing loss (HL) is frequently associated with an abnormal inner ear. To determine whether inner ear abnormalities underlie HL in $SOX10^{+/R109W}$ pigs, we

**Table 1.** Screening results.

| Dominant screen | | Total | Heritable mutant lines |
|---|---|---|---|
| | Pigs screened for dysmorphology phenotypes | **6,770** | - |
| | Tested for blood-based screens | 2,947 | – |
| | Genetic confirmation crosses | 36 | – |
| | Confirmation crosses still in progress | 7 | – |
| | Successfully confirmed crosses/mutants | 7 | – |
| **Recessive screen** | | | |
| | G3 pigs screened | 6,825 | – |
| | Confirmation crosses | 91 | – |
| | Confirmation crosses still in progress | 67 | – |
| | Successfully confirmed crosses/mutants | 22 | – |
| | | | |
| **Growth or weight defects** | | | |
| | Hind-limb paralysis | 1 | 1 |
| | Large body weight | 1 | 1 |
| | Small body weight | 15 | 2 |
| **Dysmorphology screen** | | | |
| | Behavioral abnormalities | | |
| | —Shaking | 1 | 1 |
| | —Skeletal defects | 4 | 2 |
| | —Developmental abnormalities | 4 | 3 |
| | —Testicular abnormalities | 2 | 1 |
| | —Abnormal genitals | 4 | 1 |
| | —Polydactyly | 1 | 1 |
| | —Curly coat | 1 | 1 |
| | —Coat color change | 24 | 8 |
| | Eye defects | | |
| | —Cataract | 1 | - |
| | —Color change | 2 | - |
| | Skin abnormalities | | |
| | —Chapped skin | 1 | 1 |
| | —Nude pig | 2 | 2 |
| | Ear defects | | |
| | —Large ears | 1 | – |
| | —Small ears | 1 | – |
| **Blood-based screen** | | | |
| | High cholesterol/low density lipoprotein | 8 | 2 |
| | Hyperglycemia | 4 | 1 |
| **Hernia** | | 6 | 1 |

examined the histology of the cochlea by celloidin embedding at birth. The results showed that the mutant miniature pig inner ear has only one and a half cochlea turns (red arrow in *Figure 4E*, upper panel), top loop fusing, a short modiolus, and stunted growth, which perfectly mimics the syndrome of human Mondini dysplasia (*Figure 4E*, upper panel). By contrast, the wild-type pig inner ear has three cochlear turns. The abnormality of the inner cochlea was further confirmed using micro CT

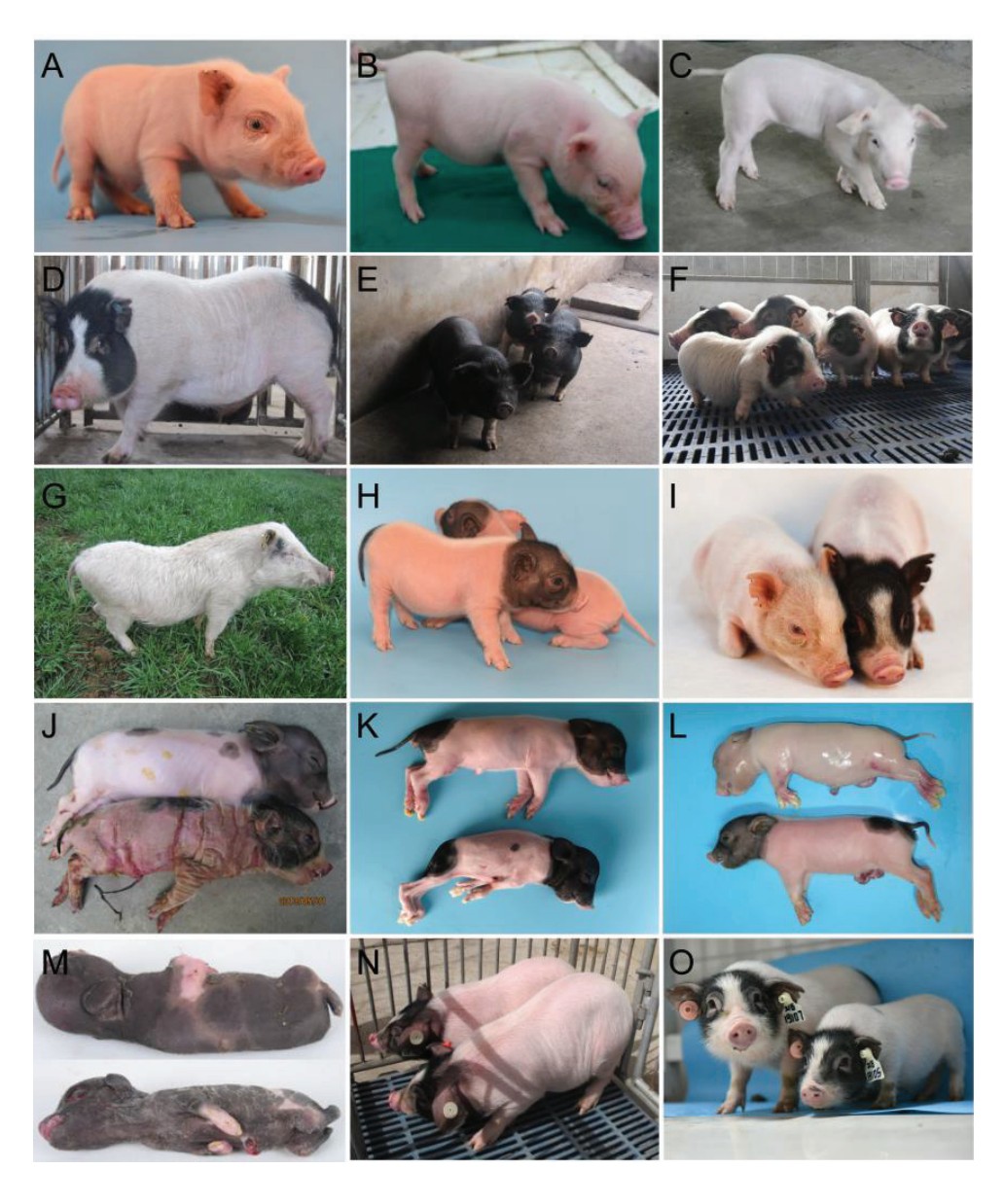

**Figure 3.** Phenotypes of confirmed dominant and recessive mutant pigs. (A) The mutant exhibited white coat color, diluted iris pigmentation and hearing loss. Histologic sections were examined from the cochleas of mutant pigs, and the striae vascularis exhibited enlarged marginal cell nuclei and tectorial membrane, indicating an abnormal organ of Corti. (B) The mutant exhibited white coat color and hearing loss. Histologic analyses of the cochleas from mutant pigs showed an extensive collapse of Reissner's membrane onto the stria vascularis and the organ of Corti. In addition, the auditory hair cells were absent or severely diminished, and the supporting cells displayed severe abnormalities. (C) The mutant was characterized by shaking and trembling, and progressive muscle weakness was also observed. (D) The mutant presented elevated blood glucose levels (hyperglycemia). Thesemutants exhibited 212.8% higher blood glucose concentrations (mutants: 8.3 mg/dL, WT: 3.9 mg/dL). (E) The mutant was identified by its black coat color. (F) the mutant showed short limbs and small body size, and this phenotype was inherited with a dominant model. (G) Line TBB007T095. The mutant was characterized by hearing loss, white coat color and diluted iris pigmentation. *SOX10* was identified as the causative gene, and this mutant may represent the first inherited animal model of human Mondini dysplasia. The mutant phenotypes shown in (A–G) exhibit dominant inheritance. (H) The mutants from three mutant lines (Z0017, Z0022 and Z0040) presented 'single-end black' coat color phenotype. Linkage analysis of line Z0040 revealed a significant signal at chr13: 49–76 Mb region. The G1 boars of lines Z0017, Z0022 and Z0040 were derived from different G0 boars, and further analysis might identify additional linkage regions in lines Z0017 and Z0022. (I) The mutant displayed diluted brown coat color (line Z0015). (J) The mutant showed ectropion, flattening of the ears and large, thick, plate-like scales over the entire body (line Z0009). (K) The mutant showed neonatal death, congenital malformations of the limbs, and a shortened lower jaw (line Z0037). (L) The mutant, presenting an autosomal recessive pattern of inheritance, displayed weak in vitality and nude skin (line Z0013). (M) The mutants from lines Z0078 and Z0079 presented abnormal facial and limb

*Figure 3 continued on next page*

Figure 3 continued

development. The G1 boars of lines Z0078 and Z0079 were derived from the same G0 boar, suggesting the same causative genes for these two mutant lines. (N) The mutant line had increased body weight and high daily weight gain that was inherited in a recessive manner (line Z0006). (O) The mutant, which is inherited with a recessive model, is associated with short limbs, small body size, and low body weight (line Z0071). The mutant phenotypes (H–O) exhibit recessive inheritance.

(*Figure 4E*, middle panel) and collodion H&E staining results (*Figure 4E*, lower panel). These data suggest that *SOX10* might be responsible for human Mondini dysplasia, and this strain represents the first inherited animal model of human dysplasia.

## Elevated expression of *FBXO32* might be responsible for congenital splay leg syndrome in pigs

Congenital splay leg syndrome is a major cause of lameness in newborn piglets and occurs at highly varying frequencies (*Dobson, 1968*). The etiology and pathogenetic mechanisms of this disease are poorly understood (*Maak et al., 2009*). Mutants of the Z0075 line exhibited hind-limb paralysis with a recessive inheritance pattern (33%, 9/27) (*Figure 5A*), with muscle atrophy and interstitial fibrosis in the hind limb skeletal muscles (semitendinosus) and longissimus dorsi but not in the fore-limb muscles (*Figure 5B*). Genetic mapping results showed the strongest linkage signal at the chr4: 16–17 Mb region, within which 20 genes with certain function are found (*Figure 5C*, *Figure 5—figure supplement 1* and *Supplementary file 4*). Interestingly, *FBXO32*, previously identified as a skeletal muscle atrophy related gene, is located in this region. Although the disease-causing mutation has not yet been located in the sequences of these genes, expression analysis indicated that *FBXO32* was highly expressed in the skeletal muscles in the affected animals at both the transcriptional and protein levels (*Figure 5D,E*). The expression of MyoD and MyoG, the master regulators of skeletal myogenesis, which drives the differentiation of myoblasts into multinucleated myotubes (*Rudnicki et al., 1993*), was downregulated in the skeletal muscles, whereas expression of another main regulator, *MYF5*, remained unchanged in these mutants, suggesting the potential negative regulation of MyoD and MyoG by *FBXO32* (*Tintignac et al., 2005*).

**Table 2.** Mapping of ENU-induced mutations.

| Line | Phenotype description | Confirmed in multiple litters | Inherited in 2nd-generation males | Chromosome regions | Affected frequencies/total offspring (%) |
|---|---|---|---|---|---|
| TBB007T095 | Hearing loss, white coat color | yes | yes | chr5: 7–71 Mb | 79/175 (45.1%) |
| Z0017, Z0022 and Z0040 | Single end black | yes | yes | chr13: 49–76 Mb (Z0040) | 6/34 (20.6%) 10/38 (26.3%) 33/99 (33.3%) |
| Z0015 | Diluted brown | yes | yes | chr15: 50–70 Mb | 14/65 (21.5%) |
| Z0009 | Chapped skin | yes | yes | chr15: 120–134 Mb | 10/50 (20.0%) |
| Z0013 | Nude skin | yes | yes | chr1: 120–144 Mb | 30/112 (26.8%) |
| Z0037 | Birth defect | yes | yes | chr2: 0–3 Mb | 19/69 (27.5%) |
| Z0078/79 | Birth defect | yes | yes | chr13: 210–216 Mb (Z0078 and Z0079) | 21/106 (19.8%) |
| Z0075 | Muscle atrophy | yes | yes | chr4: 16–17 Mb | 9/27 (33.3%) |
| Z0006 | Increased growth rate | yes | yes | chr6: 122–123 Mb | 14/60 (23.3%) |
| Z0071 | Short limbs | yes | yes | chr18: 40–50 Mb | 14/29 (48.3%) |

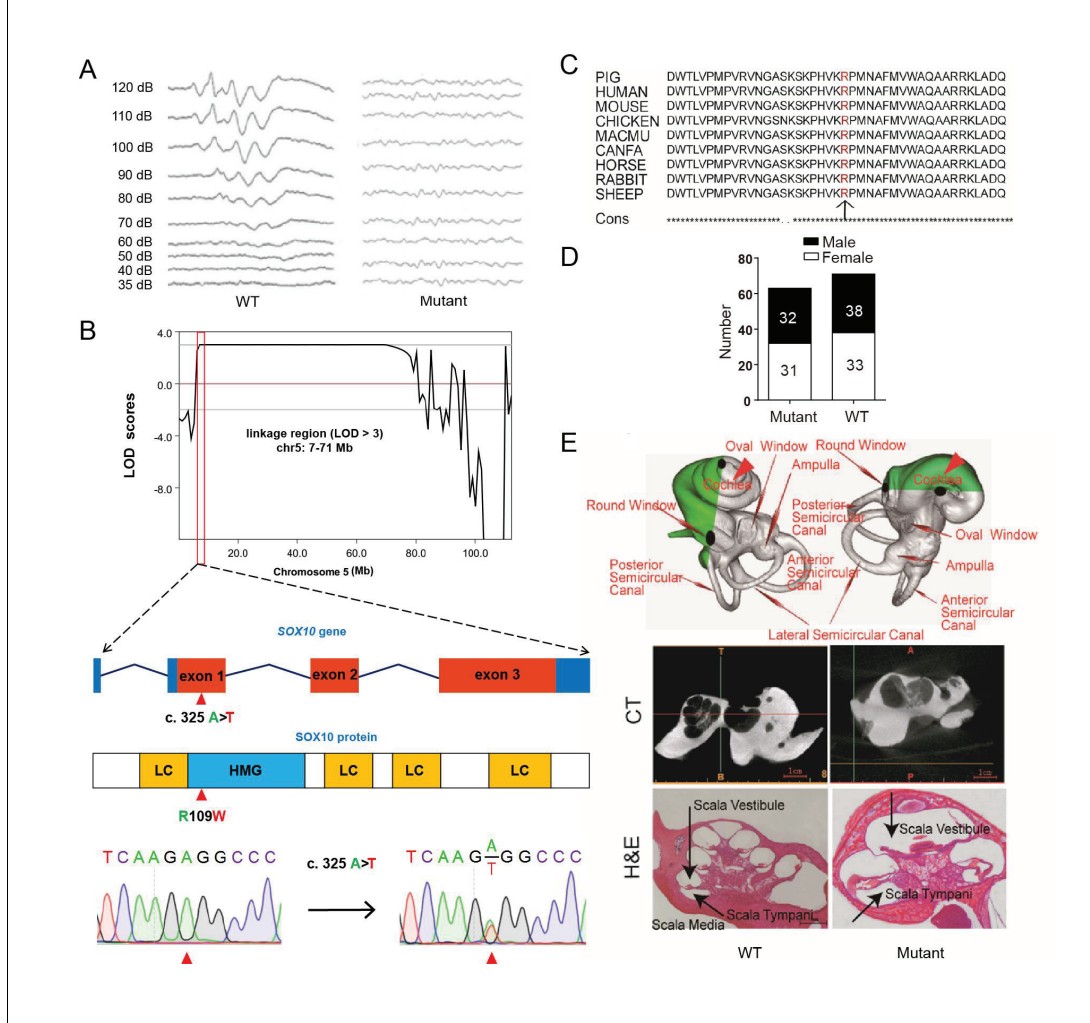

**Figure 4.** A miniature pig model of human Mondini dysplasia. (**A**) ABR tests showed profound hearing loss in both the left and right ears of mutant pigs at postnatal day 7. (**B**) GWLS results showed significant linkage at the chr5: 7–71 Mb region, within which a causative mutation (c. 325 A>T) in the exon 1 of *SOX10* locus was identified. This mutation may disrupt the *SOX10* HMG domain, which is involved in DNA binding and protein–protein interactions. HMG, high mobility group; LC, low complexity regions. (**C**) The mutation resulted in a single amino acid change (R109W) at a highly conservative region and was predicted to have a deleterious effect on gene function using SIFT software. (**D**) The causative mutation completely co-segregated with the phenotype of mutant pigs in the TBB007T095 pedigree, and the ratio of wild-type and mutant pigs confirmed that the hearing loss syndrome had an autosomal dominant inheritance pattern. (**E**) Representative images of the inner ear of the wild-type and mutant pigs. Upper panel: micro-CT 3D reconstruction results showed that the mutant had a hypoplastic inner ear malformation and fewer coils (1.5 vs. 3.5) in the cochlea, which precisely mimics the syndrome of human Mondini dysplasia. Middle panel: CT scans of the miniature pig temporal bone showed visible cochlear tip structure in the wild-type and mutant miniature pigs. Lower panel: collodion H&E staining of the wild-type pig cochlea showed three and a half turns and structural integrity of Corti's organ within the cochlear partition, which is similar to the human inner ear. Cochlea in the mutant showed fewer cochlear turns, a short modiolus, an incomplete cochlear partition, and a visible bottom of Corti's organ and spiral ganglion structure. Scale bar: 1 mm.

The following source data and figure supplements are available for figure 4:

**Source data 1.** Genome-wide parametric LOD score analyses for white coat color and hearing-loss mutant pedigree.

**Figure supplement 1.** Genome-wide linkage analysis in the white coat color and hearing loss mutant pedigree.

**Figure supplement 2.** Identification of causative mutation by a filtering procedure.

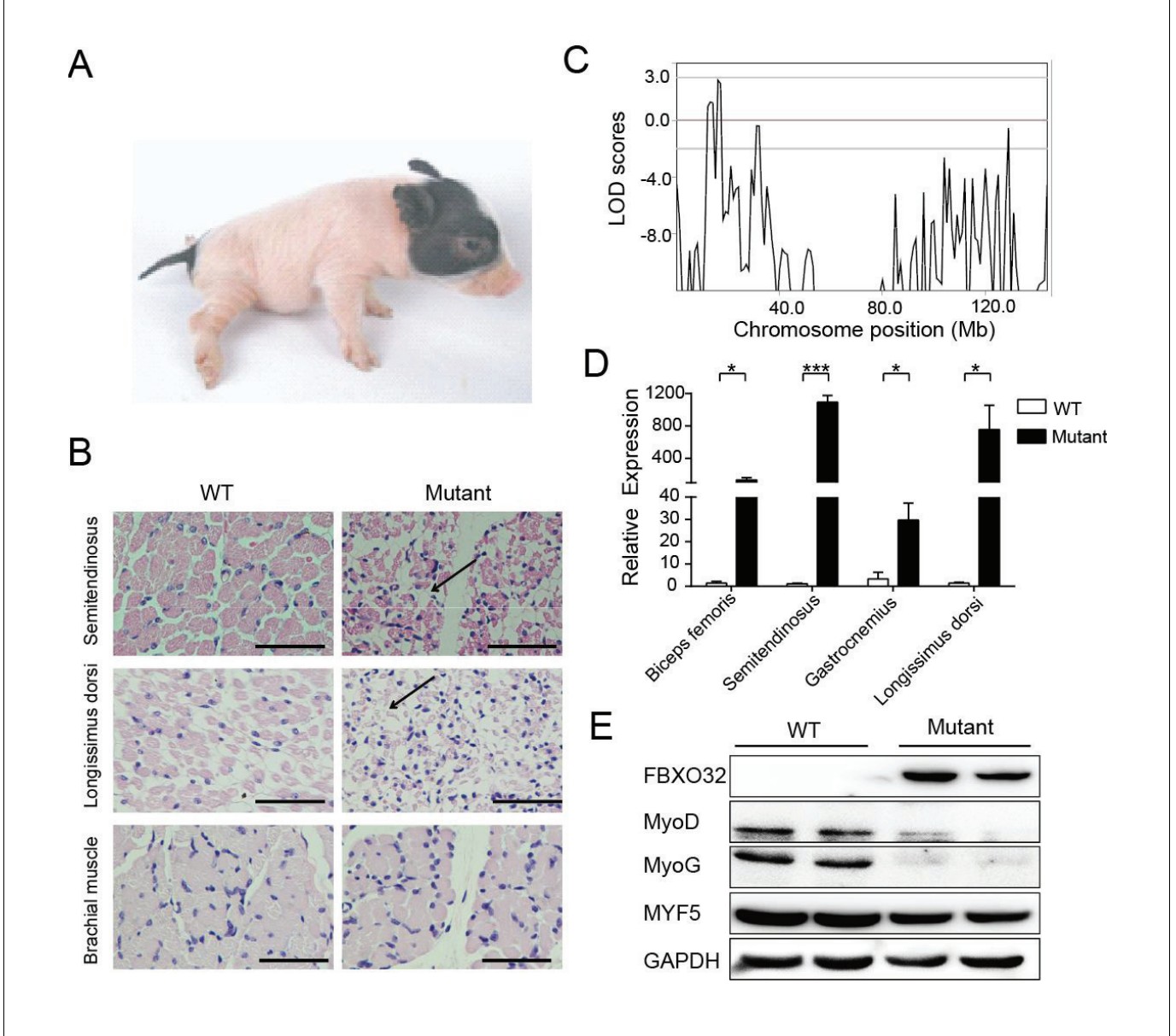

**Figure 5.** A mutant family with congenital splay leg syndrome (Z0075). (**A**) An affected piglet with the splay hind leg. (**B**) H&E staining of the semitendinosus and longissimus dorsi showing the presence of interstitial fibrosis and the absence of forelimb muscle (brachial muscle) in the mutants (arrow). Scale bar: 50 µm. (**C**) GWLS analysis showing the disease locus mapped to a 16–17 Mb region of chromosome 4, where *FBXO32* was included. (**D**) qPCR results showed that *FBXO32* was highly upregulated in the skeletal muscle of the affected animals (n = 3) compared to normal controls (n = 3). *p<0.05, ***p<0.001 by unpaired Student's t test. (**E**) Western blot results showed that FBXO32 was upregulated, and that MyoD and MyoG, but not MYF5 the master regulator of skeletal myogenesis, were downregulated in the affected piglets. GAPDH was used as a loading control.

The following source data and figure supplement are available for figure 5:

**Source data 1.** Genome-wide parametric LOD score analyses for the splay-leg mutant pedigree.

**Figure supplement 5.** Genome-wide linkage analysis in the splay-leg mutant pedigree.

## Discussion

Animal models have played crucial roles in providing our understanding of disease pathogenesis and in developing novel therapeutic agents and treatments (*McGonigle and Ruggeri, 2014*). The ideal animal model should resemble both a human disease phenotype and its underlying causality. Although inbred and genetically modified strains of classical model organisms, such as *C. elegans*, fruit flies, zebrafish and rodents, have generated many experimental data and have provided information that has helped us to understand human biology and diseases, it has become increasingly clear that many of these organisms are imperfect representations of the complex, diverse and multifaceted spectrum of human biology, leading to a demand for alternative models (*Bendixen et al., 2010*; *Prabhakar, 2012*). Indeed, the drawbacks of the conventional model organisms have hindered the direct applicability of model-obtained knowledge to human therapies, and the translation of preclinical studies in rodents to clinical trials in humans is frequently unsuccessful (*Abarbanell et al., 2010*; *Wall and Shani, 2008*).

Owing to the similarities in organ size, anatomy, physiology, metabolism, neurobiology and genome between pigs and humans, pigs are increasingly used as novel model animals in biomedical research (*Yao et al., 2016*). Pigs could be used as alternatives to dogs or monkeys as the non-rodent species in preclinical toxicological testing of pharmaceuticals (*Swindle et al., 2012*). Over the past two decades, pigs also replaced dogs as surgical models in the international arena for both training and research. However, the progress of pig genomics and functional genomics is lagging behind in health research. For example, the swine genome sequence is on version 10.2, whereas the human and mouse genomes are on versions 37 and 38, respectively (*Prather, 2013*).

In light of this background, we conducted a pilot forward genetics study to determine the potential for the use of ENU mutagenesis in pigs to meet the requirements of biomedical research and agricultural production. As a point mutagen, ENU produces not only null alleles but also hypomorphs and gain of function mutations, which are more similar to many mutations associated with human diseases (*Wilson et al., 2005*). Hypomorphic phenotypes or partial loss of function can be significantly different from the null phenotype normally produced by knockout approaches and can provide information on the function of genes in developmental or pathological processes. Correspondingly, several mutants have been identified in our screen that may serve as models for human diseases, such as hearing loss and metabolic syndromes, thus providing valuable materials for deciphering the genetic architecture of pig production traits.

As there is no reference that describes optimized ENU mutagenesis in pigs, we optimized the ENU treatment protocol in pigs and observed that dosing with three injections of 85 mg/kg at 1 week intervals is effective to introduce mutations in the pig genome at a frequency ($5.86 \times 10^{-6}$) similar to that observed in mice (*Concepcion et al., 2004*). The average rate of de novo mutation is approximately $1.2 \times 10^{-8}$ mutations per site per generation in humans (*Kong et al., 2012*; *Roach et al., 2010*). Thus, the results of the present study indicated that ENU can significantly increase the mutation rate by 130–480-fold in pigs when using the current protocol. Taking the predicted coding sequence size of the pig genome, around 4,580–16,466 ENU-induced mutations could be identified per G1 pig across the whole genome, and around 57–205 variants might occur in the coding regions. Based on ENU mutagenesis in mice, assuming that ~55% of the ENU-induced mutations in coding sequences result in amino acid changes (*Quwailid et al., 2004*), approximately 31–113 functional mutations might be detected in each G1 pig. Approximately 710 G1 pigs are required to identify one functional mutation in a specific gene from the pig genome (the number of coding genes in pig is $\sim 2.2 \times 10^4$, and the average length of the coding DNA sequence (CDS) of a gene is $1.6 \times 10^3$ bp). In mice, the optimal dose produces approximately one mutation with phenotypic effects per gene per 700–1000 gametes (*Hitotsumachi et al., 1985*). Therefore, it is feasible to establish gene-driven mutagenesis by archiving sufficient DNA and sperm from G1 pigs. Together with efficient screening protocols, this approach will provide an efficient platform for the generation of any type of allelic variation and missense or null mutations (knockout).

One of the major rate-limiting steps in ENU mutagenesis is the identification of individual causative mutations (*Hrabé de Angelis et al., 2000*). For pig mutagenesis, considering the length of gestation, the conventional mapping process is a significant bottleneck. However, with the rapidly decreasing cost of whole-genome sequencing, all G3 ENU pigs can be sequenced at a depth

sufficient to identify and segregate all of the mutations for linkage analysis (*Bull et al., 2013*). Using this strategy, 10 mutant lines were successfully used to map causative mutations or regions.

The screen pipeline described in the present study primarily focuses on visible defects (e.g., coat color and dysmorphology), blood-based chemical measurement and growth performance-related traits. The identified inheritable mutants showed a broad range of phenotypes of developmental abnormalities, including hearing loss, skeletal defects, coat color changes, hind-limb paralysis, nude and chapped skin, and growth rate, which could be potentially developed into models for human disease and dissecting the underlying mechanism of agricultural production traits.

Dominant hearing-loss models (TBB007T095) were identified and characterized as deafness with cochlear abnormalities, which mimics the pathology of the hearing loss disease Mondini dysplasia. Hearing loss is the most common sensory disorder, and congenital genetic deafness has an incidence of 1 in 1,600 children (*Iizuka et al., 2015*). Animal models of human genetic deafness play a central role in the exploration and development of new therapies (*Vodicka et al., 2005*), but rodent models cannot satisfy the demand due to the smaller size of their cochlear organs and the physiology of rodent hearing. However, human hearing and hearing in pigs have many physiological similarities, and the breeding and handling advantages of pigs make pigs the most suitable animal model for clinical otological and audiological studies. Mondini dysplasia occurs at a rate of 0.5–1/1000 in children and can be identified in various syndromes (*Deenadayal et al., 2010*). The major pathologies of Mondini dysplasia are cochlear malformation with dilatation of the vestibule, aqueduct, and ampulla and the incomplete partitioning of the cochlea (*Zheng et al., 2002*). The genetic mechanism and inheritance pattern of Mondini dysplasia are unknown, and no human inheritance family or animal model has been reported. The affected pigs in the TBB007T095 strain resemble the syndrome corresponding to human Mondini dysplasia and represent the first animal model for this disease. The mapping results for this mutant line also implied that *SOX10* might be responsible for human Mondini dysplasia. This finding provides a new clue in identifying the underlying mechanism of Mondini dysplasia, clearly highlighting the advantage of forward genetics strategies.

Unraveling the genetic composition of economic traits is a major goal in modern animal genetics and breeding. A large number of quantitative trait loci (QTL) have been mapped for a wide range of economically important traits in pigs (*Ernst and Steibel, 2013*). However, only limited numbers of QTL were mapped to defined quantitative trait genes (QTGs) and quantitative trait nucleotides (QTNs), which can be used for direct selection in pig breeding. A mutant line (Z0006) was identified with increased body weight (50% higher) and high daily gain (compared with its littermates) that were inherited in a recessive fashion . Whole-genome association analysis showed that chromosome 6 had a significant association (p<0.001) with the large body weight phenotype. Three other strains (Z0139, Z0154 and Z0156) were identified with decreased body size and weight, providing the potential for breeding miniaturized laboratory pigs with adult body weights of less than 10 kg or for identifying growth-related genes. Understanding the genetic basis of these mutants will contribute to the selection and improvement of pig economic traits.

Splay leg is the most frequently observed hereditary disorder in newborn piglets, with prevalence ranging from less than 1% to greater than 8% in certain farms (*Partlow et al., 1993*). The affected piglets are characterized by an impaired ability to stand and walk, resulting from a muscular weakness of the hind limbs (*Thurley et al., 1967*). More than 50% of affected piglets will die, making congenital splay leg a source of considerable economic losses in pig production (*Dobson, 1968*). To date, the pathogenesis and etiology of this disease are poorly understood (*Maak et al., 2009*), which impairs the elimination of these birth defects. Atrophy marker *FBXO32* is one of the putative causative genes and showed increased expression in the skeletal muscles of affected piglets (*Ooi et al., 2006*), but the role of *FBXO32* in pig congenital splay leg syndrome is neverthless controversial (*Boettcher et al., 2008*). *Maak et al. (2009)* couldn't confirm the role of *FBXO32* in pig muscle atrophy and observed a large individual variability in *FBXO32* expression in the skeletal muscles of splay-leg piglets, which has also been described previously (*Björklund et al., 1987*; *Boettcher et al., 2008*). In the present study, the Z0075 line exhibited hind-limb paralysis in a recessive inheritance pattern at a rate of 9/27 (33%), and *FBXO32* was highly expressed in the skeletal muscle of all the affected animals. In addition, MyoD and MyoG, two main regulators of skeletal muscle development (*Rudnicki et al., 1993*), were downregulated in these mutants, implying the potential negative regulation of *FBXO32* on myogenesis factors (*Tintignac et al., 2005*); however, normal levels of Myf5 may explain the presence of myogenesis in mutants (*Rudnicki et al., 1993*).

Therefore, we provided more evidence that *FBXO32* might be responsible for congenital splay-leg syndrome in pigs, thereby enhancing the current understanding of the molecular mechanisms regulating pig muscle atrophy and remolding.

In conclusion, this pioneering research provides proof that ENU mutagenesis can be used to generate mutants efficiently and to discover genes with novel functions in pigs. The data obtained in the present study showed that genome-wide ENU mutagenesis and subsequent phenotypic screening can be used to generate pig mutants with a broad range of phenotypes. This contributes to our understanding of pig genetics and provides both valuable human disease models and materials for pig breeding. In combination with gene-targeting strategies, ENU mutagenesis will continue to play an important role in the identification and characterization of mutations that underlie human diseases and pig economic traits. Based on the results of the present study, the importance of pigs as a biomedical model will further increase in the near future.

# Materials and methods

### Animals

The Bama miniatures used in this study had ad libitum access to a commercial pig diet (nutrient levels according to the United States National Research Council) and water throughout the experimental period. All experiments involving animals were performed according to the protocols approved by the Institutional Animal Care and Use Committee of the Institute of Zoology, Chinese Academy of Sciences, China.

### ENU treatment and mutagenesis

Bama miniatures (G0) were injected with ENU intravenously at a dose of 65 mg/kg or 85 mg/kg at 1 week intervals for three consecutive weeks to generate random genome-wide mutations in germ cells. A total of 82 ENU-treated G0 boars were mated with wild-type sows to produce G1 progeny. For the dominant phenotypes found in G1, at least three more wild-type females were crossed to reproduce the phenotype for inheritance testing. For the recessive phenotypes found in the G3 progenies, either the parents or the half siblings were mated for inheritance testing.

### Spectrum of ENU-induced mutations in pigs

To evaluate the mutational pattern and frequency of induced nucleotide changes in ENU mutagenesis from a genome-wide level, the trio-based 2b-RAD method was used. The 2b-RAD libraries were constructed according to the protocol developed by *Wang et al. (2012)*. A reduced representative library using adaptors with 5′-NNA-3′ and 5′-NNT-3′ overhangs was separately generated for each individual. These libraries were barcoded, pooled, and subjected to single-end sequencing (1 × 36 bp) on the Illumina HiSeq 2000 platform. The resulting raw sequencing reads were first trimmed to remove adaptor sequences and were then subjected to quality assessment using defined thresholds. After the poor-quality reads were removed, the remaining reads were mapped to the pig build 10.2 reference sequence (SusScrofa Build 10.2) using the Burrows-Wheeler aligner (BWA) tools (RRID: SCR_010910) with the default parameters. After alignment, mapped reads were used to call variants via the SAMtools toolkits. We used the SnpEff program (RRID:SCR_005191) for variant annotation, effect prediction and variant categorization. To discover the ENU-induced mutations, each family was divided into five trios (father-mother-child). The DeNovoGear tool (RRID:SCR_000670) was used to detect de novo mutations using trio sequencing data. A stringent set of criteria was adopted to reduce the false-positive rate: (a) the posterior probability of being a de novo mutation (pp_dnm) should be greater than 0.9; (b) the posterior probability of a Mendelian inheritance mutation (pp_null) should be less than 0.001; (c) de novo mutations found in more than one trio were excluded (ENU-induced mutations in different sperm cells were assumed to distribute randomly, and the possibility that G1 carried the same ENU-induced mutations was extremely low); and d) genomic sites covered by at least 10 reads in either member of the trio could be used to identify de novo mutations.

## Screening procedure

A large-scale screening pipeline was modified from RIKEN (Institute of Physical and Chemical Research) SHIRPA for mouse phenotyping (http://ja.brc.riken.jp/lab/jmc/shirpa/). Briefly, G1 and G3 Bama miniatures were screened for dysmorphology, growth traits, body size, behavior, and hearing and visual acuity at different ages of development. Dysmorphology was screened at birth and 2 months; growth traits and body size were screened at birth, day 15, 1 month, 2 months, 3 months, 4 months, 5 months and 6 months; the clinical biochemical analysis was performed at 3 months and 6 months; hearing and visual acuity was tested at birth; and behavior abnormality was monitored throughout the breeding procedure.

## Clinical biochemistry

Blood samples were obtained by puncturing the precaval vein of 3- and 6-month-old G1 and G3 animals that had been fasted overnight. Plasma from clot activator-treated blood was analyzed using a Hitachi 7080 automatic biochemical analyzer (Hitachi Instrument Ltd., Tokyo, Japan) and adapted reagents (Chemclin Biotech Co., Ltd., Beijing, China). The biochemical parameters relating to the cardiovascular system and metabolic diseases included: creatine kinase (CK), creatine kinase-MB (CK-MB), lactate dehydrogenase (LDH), alpha-hydroxybutyrate dehydrogenase (α-HBDH), high-sensitivity C-reactive protein (hs-CRP), glucose (GLU), triglycerides (TG), total cholesterol (TC), low-density lipoprotein cholesterol (LDL-c), high-density lipoprotein cholesterol (HDL-c), creatinine (CRE), blood urea nitrogen (BUN), alanine aminotransferase (ALT), aspartate transaminase (AST), and gamma-glutamyltranspeptidase (GGT).

Serum was separated by refrigerated centrifugation (2000 rpm for 5 min), and the 18 hematologic parameters were analyzed using a RAYTO RT-7600S automatic hematology analyzer (Rayto Life and Analytical Sciences Co., Ltd., Guangzhou, China). The 95% range of the values was defined as physiological when the outlier data were eliminated. The cut-off level was identified based on a value exceeding 2.0 standard deviations in both sexes of the G1 and G3 offspring. In the screening procedure for clinical biochemistry, in the cases where altered values (deviating from the cutoff level) were detected, confirmation was performed after a 2-week interval. After that, the measurements of biochemical parameter deviants were repeated at least three times in the same sample.

## Dysmorphology screen

The dysmorphology screen aimed to identify mutants with bone- and cartilage-related phenotypes as well as morphological abnormalities. In this study, a detailed procedure was established for the quick and efficient whole-body assessment of anatomical abnormalities and defects in different organ systems. These screening tests were performed from birth to 6 months of age. We designed screening parameters to detect mutants with birth defects, behavior abnormalities or defects in organ development, including bone, skin, coat color, head, ear, eye, tooth, hand, foot, limb, vagina, and testis. Through a battery of screens, we performed comprehensive monitoring of each pig and obtained a detailed phenotypic characterization.

## Growth traits and body size screen

For each pig, the body weight, body length, body depth, hip width and chest circumference were measured at the following ages: birth, day 15, 1 month, 3 months, and 6 months. The cut-off level was identified based on a value exceeding 2.0 standard deviations in G1 and G3 offspring for both sexes.

## Cryopreservation of spermatozoa, tissues and cells

Cryopreservation of mutant boar spermatozoa was conducted following the Minitube semen cryopreservation protocol. Briefly, semen was collected and analyzed for quality using a computer-assisted semen analyzer (Hamilton Inc, Beverly, MA, USA) and was diluted using Androhep Plus (Minitube Inc, Verona, WI, USA). The diluted semen was centrifuged and re-suspended with Androhep-CryoGuard and cooled to half of the final freezing volume according to total sperm counts. The same volume of CryoGuard Freezing was added to the diluted semen at 5°C. The semen suspension was then loaded in 5 ml cryopreservation straws and frozen according to a gradient cooling procedure. For thawing, the frozen straws were removed from liquid nitrogen, put into a water bath, and

maintained at 37°C for less than 5 min. Tissues were collected at FBS and snap frozen in liquid nitrogen. After ear fibroblast cells become confluent, they were frozen in liquid nitrogen for long-term storage.

## Mapping and identification of causative mutations in ENU mutants

Genomic DNA was isolated from ear tissues using a routine phenol/chloroform extraction, and whole-genome SNP genotyping was performed using porcine SNP60 BeadChips (Illumina Inc., USA) containing 62,163 SNP markers at Beijing Compass Biotecnology Co., Ltd. A family-based GWLS was used to map chromosome regions co-segregating with a phenotype within a mutant pedigree, and WGS was further used to detect the ENU-induced mutations in multiple affected pigs. In GWLS, raw data were processed by removing SNPs with a call rate <90% and a minor allele frequency (MAF) <0.05. Then, Merlin software (RRID:SCR_009289) was used to perform genome-wide and family-based linkage analysis. In addition, parametric linkage analysis assuming either dominant or recessive models was conducted, and the LOD score was calculated to assess the evidence for linkage.

In each pedigree, WGS was performed for one parental pig and two affected offspring at the Shanghai Personal Bio. Ltd, Shanghai. DNA fragments with an approximate size of 300 bp were purified and ligated to Illumina sequencing adaptors to build a sequencing library. Each pig was sequenced in a single lane of the Illumina HiSeq 2000 sequencer following the manufacturer's protocols. After poor-quality reads were removed, the remaining reads were mapped to the pig build 10.2 reference sequence using Burrows-Wheeler aligner (BWA) tools (RRID:SCR_010910). Mapped reads were used to call variants via the SAMtools (RRID:SCR_002105) and VarScan toolkits (RRID: SCR_006849). The snpEFF software (RRID:SCR_005191) was applied to predict the functional effects of the detected variants.

## CT scanning

CT scanning was performed with a 64-channel helical CT system (Lightspeed VCT LS Advantage 64 slices, GE, USA) using the following parameters: 120 kV, 225 mA, 0.5 pitch, 1 s rotation time, 0.625 mm slice thickness, 0.6 mm collimation, 512 × 512 matrix size, FOV of 16 cm, bone reconstruction algorithm. At this collimation, an isotropic voxel measuring 0.6 mm on each side was obtained. The images were acquired parallel to the orbitomeatal axial plane and were reconstructed in the coronal plane.

## Histopathology

Tissues were fixed with 4% neutral buffered formalin. The fixed tissues were embedded in paraffin according to standard laboratory procedures. 5-μm-thick sections of paraffin-embedded tissues were processed for hematoxylin-eosin (H&E) staining.

## Quantitative reverse transcriptase PCR (qRT-PCR)

Total RNA was isolated from tissues using Trizol (Thermo Fisher Scientific Inc., Carlsbad, CA, USA) and reverse transcribed to cDNA using FastQuant RT Kit (Tiangen Bio Inc, Beijing, China). qRT-PCR was performed on a Stratagene Mx3005P real-Time PCR system (Agilent Technologies Inc., Santa Clara, CA, USA) with SYBR Premix EX Taq kit (Takara Bio Inc., Otsu, Shiga, Japan). The relative mRNA expression levels of the target genes were calculated as the fold changes of the threshold cycle (Ct) value relative to the reference using the $2^{-\Delta\Delta Ct}$ method. The following primers were used:

GAPDH forward: 5'-GCAAAGTGGACATTGTCGCCATCA-3', reverse: 5'-AGCTTCCCATTCTCAGCCTTGACT-3';

*FBXO32* forward: 5'-AAGCGCTTCCTGGATGAGAA-3', reverse: 5'-GGCCGCAACATCATAGTTCA-3'.

## Western blot

Tissue proteins were lysed in RIPA lysis buffer (Cwbio Inc., Beijing, China) and a protease inhibitor cocktail (Thermo Fisher Scientific Inc., Rockford, IL, USA). Protein extracts were electrophoresed on 15% SDS-PAGE gels and transferred onto PVDF membranes (GE Healthcare Inc., Menlo Park, California, USA). Membranes were blocked with 5% skimmed milk for 2 hr at room temperature and incubated overnight at 4°C with primary antibodies against *FBXO32* (RRID: AB_2246982, Santa Cruz

Biotechnology Inc., Santa Cruz, CA, USA, 1:1000 dilution), Myod (RRID: AB_631992, Santa Cruz Biotechnology Inc., Santa Cruz, CA, USA, 1:500 dilution), MyoG (RRID: AB_784707, Santa Cruz Biotechnology Inc., Santa Cruz, CA, USA, 1:1000 dilution), MYF5 (RRID: AB_10975611, Abcam Inc., Cambridge, MA, USA, 1:1000 dilution) and GAPDH (RRID: AB_2651183, CwBio Inc., Beijing, China, 1:1000 dilution). The detection was carried out with horseradish peroxidase (HRP)–conjugated goat anti-rabbit or anti-mouse secondary antibody (ZSGB-Bio Inc., Beijing, China, 1:5000 dilution) and visualized with an enhanced chemoluminescence (ECL) kit (Thermo Fisher Scientific Inc., Rockford, IL, USA), according to the manufacturer's protocol. Each experiment was performed in triplicate.

## Statistical analysis

The statistical analysis of the data was conducted using JMP (SAS Inc., USA). Values are presented as medians and at 95% or 90% ranges unless stated otherwise. Statistical significance (defined as $p < 0.05$) was evaluated using the $\chi^2$ test.

## Acknowledgements

The authors thank all the personnel at the Beijing Farm Animals Research Center, the Chinese Academy of Sciences and the Chinese Swine Mutagenesis Consortium whose names are not listed as co-authors of this paper for their assistance. The authors also thank John R Speakman (Institute of Genetics and Developmental Biology, CAS) for useful comments on this manuscript. The study was supported by the Strategic Priority Research Program of CAS (XDA08000000 and XDA01030400), the National Transgenic Project of China (2016ZX08009003-006-007), the National Basic Research Program of China (2011CBA0100, 2011CB944100, 2011BAI15B02, 2012BAI39B04, 2013ZX08009003, and 2012CB967900), the National High Technology Research and Development Program of China (2012AA020602) and the National Natural Science Foundation of China (81671274, 81670941, and 81570933).

## Additional information

### Funding

| Funder | Grant reference number | Author |
|---|---|---|
| National Natural Science Foundation of China | 81570933 | Haitao Shang |
| Ministry of Science and Technology of the People's Republic of China | National Basic Research Program of China (2012BAI39B04) | Menghua Li |
| National Natural Science Foundation of China | 81670941 | Yong Wang |
| Ministry of agriculture of the People's Republic of China | National Transgenic Project of China (2016ZX08009003-006-007) | Hongmei Wang |
| Ministry of Science and Technology of the People's Republic of China | National Basic Research Program of China (2011CB94410) | Hongmei Wang |
| Ministry of Science and Technology of the People's Republic of China | National Basic Research Program of China (2013ZX08009003) | Hongmei Wang |
| Ministry of Science and Technology of the People's Republic of China | National Basic Research Program of China (2011BAI15B02) | Hongmei Wang |
| Ministry of Science and Technology of the People's Republic of China | National Basic Research Program of China (2012CB967900) | Shiming Yang |
| Ministry of Science and Technology of the People's Republic of China | National Basic Research Program of China (2011CBA0100) | Jianguo Zhao |

| Ministry of Science and Technology of the People's Republic of China | National High Technology Research and Development Program of China (2012AA020602) | Jianguo Zhao |
|---|---|---|
| Chinese Academy of Sciences | Strategic Priority Research Program of CAS (XDA08000000) | Jianguo Zhao |
| National Natural Science Foundation of China | 81671274 | Jianguo Zhao |
| Chinese Academy of Sciences | Strategic Priority Research Program of CAS (XDA01030400) | Jianguo Zhao |

The funders had no role in study design, data collection and interpretation, or the decision to submit the work for publication.

## Author contributions

TH, Data curation, Formal analysis, developed the protocol for the ENU treatment in pigs; CC, Data curation, Software, Formal analysis, Methodology, Writing—original draft, Writing—review and editing, conducted the bioinformatics analyses (causative gene mapping); HS, Resources, Data curation, Formal analysis, carried out the screening of large number mutants; WGu, Data curation, Formal analysis, Methodology, characterized the pig hearing loss model; YM, Data curation, Formal analysis, Supervision, conducted the mutant screening and mutangenesis banking archiving; SYan, XWe, ZY, Formal analysis, conducted the mutant screening and mutangenesis banking archiving; YZha, Data curation, Formal analysis, conducted the mutant screening; QZhe, TZ, YLi, KL, MQ, QH, RZ, XWan, QJ, XWa, GQ, YLi, WJ, JYa, JH, HZh, XX, XZ, KG, QW, SZ, LL, FX, YZha, KH, YC, PZ, HZo, LX, JX, JR, WZ, JYu, ZLi, Formal analysis, conducted the mutant screening; XWan, Formal analysis, Methodology, conducted the mutant screening; QK, HL, MLi, Formal analysis; DW, Formal analysis, conducted the mutangenesis; AL, Formal analysis, characterized the pig hearing loss model; MLi, Data curation, conducted the mutant screening; WGu, Formal analysis, Supervision; YW, Investigation; HWa, ZLi, Supervision, Investigation, Project administration; SYan, Supervision, Project administration; HWe, Conceptualization, Supervision, Funding acquisition, Project administration; JZ, Supervision, Funding acquisition, Investigation, Writing—original draft, Project administration, Writing—review and editing; QZho, Conceptualization, Supervision, Investigation, Project administration, Writing—review and editing; AM, Conceptualization, Supervision, Project administration, Writing—review and editing

## Author ORCIDs

Tang Hai, http://orcid.org/0000-0002-3192-0753
Chunwei Cao, http://orcid.org/0000-0001-8114-7589
Jianguo Zhao, http://orcid.org/0000-0001-6587-4823

## Ethics

Animal experimentation: This study was performed in strict accordance with the recommendations in the Guide for the Care and Use of Laboratory Animals of the National Institutes of Health. All experiments involving animals were performed according to the protocols approved by the Institutional Animal Care and Use Committee of the Institute of Zoology, Chinese Academy of Sciences, China. All surgery was performed under sodium pentobarbital anesthesia, and every effort was made to minimize suffering.

# Additional files

## Supplementary files

• Supplementary file 1. Summary of the 2b-RAD sequencing analysis.
• Supplementary file 2. Summary of the sequenced ENU-induced mutations in two families.

• Supplementary file 3. The frequency and distribution of *SOX10* c. 325 A>T mutation among TBB007T095 lines and other breeds.

• Supplementary file 4. Genes located in the linkage region of chr4: 16–17 Mb.

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
