## [Decision Letter]

Thank you for submitting your article "Large-scale production of mutant pigs by ENU mutagenesis: A pilot study" for consideration by *eLife*. Your article has been reviewed by two peer reviewers, one of whom, Hong Zhang (Reviewer #1), is a member of our Board of Reviewing Editors and the evaluation has been overseen by Marianne Bronner as the Senior Editor. The following individuals involved in review of your submission have agreed to reveal their identity: Mary Mullins (Reviewer #3).

The reviewers have discussed the reviews with one another and the Reviewing Editor has drafted this decision to help you prepare a revised submission.

This is an impressive scientific endeavor to perform ENU mutagenesis and a forward genetic screen in a miniature pig species. This large collaborative effort demonstrates in this pilot screen the feasibility and value of performing such a screen in a larger mammal like the pig, providing multiple potential models for human disease and animal traits.

Overall there is too brief a presentation of the results, so the study cannot be fully appreciated by the reader or fully evaluated. Further documentation of the results is needed in the paper. Much more clarity and caution needs to be applied to the description of experiments, results and, most importantly, conclusions. These concerns can be addressed by significant re-writing.

1) The mutant with hearing loss was found to have a linked mutation in the Sox10 gene. The linkage analysis should be shown, often this is done by showing linkage scores throughout the genome. Without additional results, the mutation found in Sox10 cannot be referred to as 'causative'. In the Discussion the authors are appropriately cautious in this respect. How many mutants and wild types were examined for linkage to this SNP and what were the results? It should be possible to assess recombination in the G3 progeny? If so, where do the recombinants map to and how many genes were found in the physical interval defined by flanking recombinants to the SNP? How deep and complete was the sequencing coverage of the trios and how complete is the pig genome sequence in this region, that is, are there gaps with genes that could be missed?

2) The Material and methods state that each family was divided into 5 mother-father-child trios. Isn't the father always the same for a particular mutation, the G1 backcrossed male? Was the mapping strategy the same for dominant and recessive mutations? Were backcrosses also done for the dominant G1 mutants? More information is needed on how the mapping was done.

3) Please explain the axes in Figure 4 and what the peaks reflect. Was linkage found anywhere else in the genome? With the n=9 mutants, there could be other possible linkage sites and confidence for the FbxO32 gene cannot be definitive.

4) For the splay leg mutant, do all muscles show the same defect as in Figure 4 or is it restricted or more severe more posteriorly, for example, in the hindlimbs compared to the forelimbs?

5) Is there a mutation in or near the FbxO32 gene? In Figure 4 are heterozygotes in the WT bar graph? It is unusual to have a mutation in a gene that causes its increased expression that is not dominant. Heterozygotes should be examined and the qPCR results included in Figure 4 also. Have the FoxO32 primer sites and sequence length been examined in the WT to ensure there are no polymorphisms that would result in the strongly reduced qPCR? The protein level difference seems less than the qPCR results between WT and mutant. It cannot be said that FBXO32 is the causative change in this mutant without further evidence. Linkage is insufficient evidence for proof of causation. No mutation, for example, was reported in this gene and FbxO32 gene is highly variably expressed generally.

6) Z009, Z0078, Z0079, Z0006, Z0139, Z0154 and Z0156, the coat color mutants and all mutant in Table 2, should be documented with the phenotypes shown in Figures.

7) Only 1 of the 7 confirmed dominant mutations is described in the text. Describe and document them all including photos, even if not mapped to a chromosome location yet, since this is the main point of the paper.

---

## [Author Response]

*1) The mutant with hearing loss was found to have a linked mutation in the Sox10 gene. The linkage analysis should be shown, often this is done by showing linkage scores throughout the genome. Without additional results, the mutation found in Sox10 cannot be referred to as 'causative'. In the Discussion the authors are appropriately cautious in this respect. How many mutants and wild types were examined for linkage to this SNP and what were the results? It should be possible to assess recombination in the G3 progeny? If so, where do the recombinants map to and how many genes were found in the physical interval defined by flanking recombinants to the SNP? How deep and complete was the sequencing coverage of the trios and how complete is the pig genome sequence in this region, that is, are there gaps with genes that could be missed?*

We appreciate the constructive suggestions from the reviewer.

To map the causative mutation, 10 mutants and 17 wild types were selected from this pedigree to perform the linkage analysis, and the linkage scores throughout the whole genome were shown in Figure 4—figure supplement 1. As indicated, across the whole genome, only one significant linkage signal (chr5:7-71 Mb, LOD>3) was detected. Within this region, 1086 genes with certain functions were found. Considering the mutants showed withe coat color, hypopigmentation related gene-*SOX10* was first selected as a candidate gene, and a missense mutation (c. 325 A>T) which resulted in a single amino acid substitution (R109W, Figure 4) was identified by sequencing the coding sequences of *SOX10*. Additionally, whole genome sequencing of two independent mutants and one control with average read-depth coverage of 11.51×, 10.64×, and 10.63× respectively, was also performed, and a procedure was designed for filtering the causative mutation. Eventually, only the mutation (c. 325 A>T) in *SOX10*, but not other mutations, were found to completely co-segregate with the phenotype of the pedigree. As shown in [Supplementary-material SD6-data], 63 mutants from this pedigree all carried AT genotypes, however, the 71 wild-types only presented AA genotypes. Furthermore, this mutation was not observed in other G1 founders and commercial pig breeds (large white, landrace, and Duroc), as shown in [Supplementary-material SD6-data]. Additionally, the *Sox10* KO mouse model also displays a lack of melanocytes and white spotting coat color (MGI database, http://www.informatics.jax.org/marker/MGI:98358) which is consistent with the current pig mutants in coat color.

Given all this evidence, we suggest that the *SOX10* mutation might be the causative mutation of this mutant phenotype, and the possibility that an additional causative mutation located in gaps of pig genome was low. The detailed information about mapping now added in the Results (subsection “A mutation in *SOX10* causes inner ear malfunctions and mimics human Mondini dysplasia” and Figure 4—figure supplement 2).

*2) The Material and methods state that each family was divided into 5 mother-father-child trios. Isn't the father always the same for a particular mutation, the G1 backcrossed male? Was the mapping strategy the same for dominant and recessive mutations? Were backcrosses also done for the dominant G1 mutants? More information is needed on how the mapping was done.*

To evaluate the mutational pattern and frequency of induced nucleotide changes in ENU mutagenesis from a genome wide level, a trio-based 2b-RAD method was applied. The details can be found in the Material and methods, subsection “Spectrum of ENU-induced mutations in pigs”. The “father” in the trio was G0 boar, and the “child” in the trio was a G1 boar. Each G1 progeny boar derived from a specific ENU mutagenized sperm of G0, and thus the 5 G1 boars in the trios are different and should carry different ENU induced mutations. To avoid confusion, this point was highlighted in the Results.

The mapping strategy was same for dominant and recessive mutations in our study. To map the ENU-induced causative mutation for dominant and recessive mutants, family-based linkage analysis and whole genome sequencing were performed. The detailed information for these procedures can be found in the Material and methods, subsection “Mapping and identification of causative mutations in ENU mutant”. We didn’t back cross the dominant G1 mutants for inheritance testing.

*3) Please explain the axes in Figure 4 and what the peaks reflect. Was linkage found anywhere else in the genome? With the n=9 mutants, there could be other possible linkage sites and confidence for the FbxO32 gene cannot be definitive.*

The axes in Figure 4 (now Figure 5) have been updated. The x-axis represented chromosome position, and y-axis represented the LOD scores.

The linkage scores throughout the whole genome were shown in Figure 5—figure supplement 1, and the strongest signal showing suggestive linkage (LOD=2.81) was found in chr4 (16-17 Mb), within which twenty genes with certain function were found (Figure 5, Figure 5—figure supplement 1 and [Supplementary-material SD7-data]). Interestingly, *FBXO32*, previously identified as a well-known skeletal muscle atrophy related gene (Bodine et al., 2001 and 2014), was located in the linkage region. While identification of the disease-causing mutation is still ongoing, the expression analysis indicated that *FBXO32* was significantly elevated in the skeletal muscles of mutant compared with wild-type pigs. We agree with the opinions from the reviewers that current study only provides preliminary data that *FBXO32* is responsible for the muscle atrophy, and further experiments are necessary to elucidate the genetic and molecular mechanism of *FBXO32* in muscle atrophy.

[2] Bodine S C, Latres E, Baumhueter S, et al. Identification of ubiquitin ligases required for skeletal muscle atrophy[J]. Science, 2001, 294(5547): 1704-1708.

[3] Bodine S C, Baehr L M. Skeletal muscle atrophy and the E3 ubiquitin ligases *Murf1* and MAFbx/atrogin-1[J]. American Journal of Physiology-Endocrinology and Metabolism, 2014, 307(6): E469-E484.

*4) For the splay leg mutant, do all muscles show the same defect as in Figure 4 or is it restricted or more severe more posteriorly, for example, in the hindlimbs compared to the forelimbs?*

As shown in the Figure 5, the defects in hind limb muscle and longissimus dorsi are more severe. However, muscles appear normal in forelimb of mutants from the H&E staining.

*5) Is there a mutation in or near the FbxO32 gene? In Figure 4 are heterozygotes in the WT bar graph? It is unusual to have a mutation in a gene that causes its increased expression that is not dominant. Heterozygotes should be examined and the qPCR results included in Figure 4 also. Have the FoxO32 primer sites and sequence length been examined in the WT to ensure there are no polymorphisms that would result in the strongly reduced qPCR? The protein level difference seems less than the qPCR results between WT and mutant. It cannot be said that FBXO32 is the causative change in this mutant without further evidence. Linkage is insufficient evidence for proof of causation. No mutation, for example, was reported in this gene and FbxO32 gene is highly variably expressed generally.*

As we described above, the mapping and screen of the causative mutation in this mutant family is still ongoing, so we could not define the heterozygotes right now. The WT in Figure 4 (now Figure 5) included just littermate piglets with wild-type phenotypes. As indicated in Figure 6 we checked the sequence of primer and PCR amplification fragments in different pigs and didn’t find any mutations which indicated the qPCR results are reliable. As we stated above, we agreed the comments from the reviewer that *FBXO32* is the candidate gene responsible for this mutant phenotype and need to be further studied. However our results provide more evidence that *FBXO32* is playing vital roles in the muscle atrophy.

Author response image 1.Sequencing results of forward and reverse primers.**DOI:**
http://dx.doi.org/10.7554/eLife.26248.020

*6) Z009, Z0078, Z0079, Z0006, Z0139, Z0154 and Z0156, the coat color mutants and all mutant in Table 2, should be documented with the phenotypes shown in Figures.*

The phenotypes in Table 2 were shown in Figure 3 to O and Figure 5.

*7) Only 1 of the 7 confirmed dominant mutations is described in the text. Describe and document them all including photos, even if not mapped to a chromosome location yet, since this is the main point of the paper.*

The phenotypes and descriptions of the confirmed dominant mutants were shown in Figure 3 to G.